# Self-Consistency Improves Chain of Thought Reasoning in Language Models

**Xuezhi Wang**[†‡]**, Jason Wei**[†]**, Dale Schuurmans**[†]**, Quoc Le**[†]**, Ed H. Chi**[†]**,
Sharan Narang**[†]**, Aakanksha Chowdhery**[†]**, Denny Zhou**[†§]

[†]Google Research, Brain Team
[‡]xuezhiw@google.com, [§]dennyzhou@google.com

## Abstract

Chain-of-thought prompting combined with pre-trained large language models has achieved encouraging results on complex reasoning tasks. In this paper, we propose a new decoding strategy, *self-consistency*, to replace the naive greedy decoding used in chain-of-thought prompting. It first samples a diverse set of reasoning paths instead of only taking the greedy one, and then selects the most consistent answer by marginalizing out the sampled reasoning paths. Self-consistency leverages the intuition that a complex reasoning problem typically admits multiple different ways of thinking leading to its unique correct answer. Our extensive empirical evaluation shows that self-consistency boosts the performance of chain-of-thought prompting with a striking margin on a range of popular arithmetic and commonsense reasoning benchmarks, including GSM8K (+17.9%), SVAMP (+11.0%), AQuA (+12.2%), StrategyQA (+6.4%) and ARC-challenge (+3.9%).

## 1 Introduction

Although language models have demonstrated remarkable success across a range of NLP tasks, their ability to demonstrate reasoning is often seen as a limitation, which cannot be overcome solely by increasing model scale (Rae et al., 2021; BIG-bench collaboration, 2021, *inter alia*). In an effort to address this shortcoming, Wei et al. (2022) have proposed *chain-of-thought prompting*, where a language model is prompted to generate a series of short sentences that mimic the reasoning process a person might employ in solving a task. For example, given the question *"If there are 3 cars in the parking lot and 2 more cars arrive, how many cars are in the parking lot?"*, instead of directly responding with *"5"*, a language model would be prompted to respond with the entire chain-of-thought: *"There are 3 cars in the parking lot already. 2 more arrive. Now there are 3 + 2 = 5 cars. The answer is 5."*. It has been observed that chain-of-thought prompting significantly improves model performance across a variety of multi-step reasoning tasks (Wei et al., 2022).

In this paper, we introduce a novel decoding strategy called *self-consistency* to replace the greedy decoding strategy used in chain-of-thought prompting (Wei et al., 2022), that further improves language models' reasoning performance by a significant margin. Self-consistency leverages the intuition that complex reasoning tasks typically admit multiple reasoning paths that reach a correct answer (Stanovich & West, 2000). The more that deliberate thinking and analysis is required for a problem (Evans, 2010), the greater the diversity of reasoning paths that can recover the answer.

Figure 1 illustrates the self-consistency method with an example. We first prompt the language model with chain-of-thought prompting, then instead of greedily decoding the optimal reasoning path, we propose a "sample-and-marginalize" decoding procedure: we first *sample* from the language model's decoder to generate a *diverse* set of reasoning paths; each reasoning path might lead to a different final answer, so we determine the optimal answer by *marginalizing out* the sampled reasoning paths to find the most consistent answer in the final answer set. Such an approach is analogous to the human experience that if multiple different ways of thinking lead to the same answer, one has greater confidence that the final answer is correct. Compared to other decoding methods, self-consistency avoids the repetitiveness and local-optimality that plague greedy decoding, while mitigating the stochasticity of a single sampled generation.

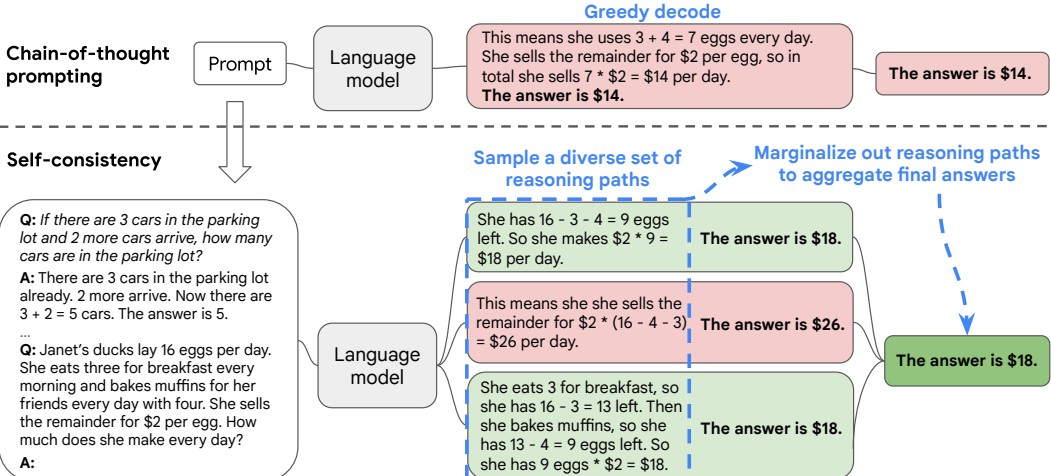

Figure 1: The self-consistency method contains three steps: (1) prompt a language model using chain-of-thought (CoT) prompting; (2) replace the "greedy decode" in CoT prompting by sampling from the language model's decoder to generate a diverse set of reasoning paths; and (3) marginalize out the reasoning paths and aggregate by choosing the most consistent answer in the final answer set.

Self-consistency is far simpler than prior approaches that either train an additional verifier (Cobbe et al., 2021) or train a re-ranker given additional human annotations to improve generation quality (Thoppilan et al., 2022). Instead, self-consistency is entirely *unsupervised*, works off-the-shelf with pre-trained language models, requires no additional human annotation, and avoids any additional training, auxiliary models or fine-tuning. Self-consistency also differs from a typical ensemble approach where multiple models are trained and the outputs from each model are aggregated, it acts more like a "self-ensemble" that works on top of a *single* language model.

We evaluate self-consistency on a wide range of arithmetic and commonsense reasoning tasks over four language models with varying scales: the public UL2-20B (Tay et al., 2022) and GPT-3-175B (Brown et al., 2020), and two densely-activated decoder-only language models: LaMDA-137B (Thoppilan et al., 2022) and PaLM-540B (Chowdhery et al., 2022). On all four language models, self-consistency improves over chain-of-thought prompting by a striking margin across all tasks. In particular, when used with PaLM-540B or GPT-3, self-consistency achieves new state-of-the-art levels of performance across arithmetic reasoning tasks, including GSM8K (Cobbe et al., 2021) (+17.9% absolute accuracy gains), SVAMP (Patel et al., 2021) (+11.0%), AQuA (Ling et al., 2017) (+12.2%), and across commonsense reasoning tasks such as StrategyQA (Geva et al., 2021) (+6.4%) and ARC-challenge (Clark et al., 2018) (+3.9%). In additional experiments, we show self-consistency can robustly boost performance on NLP tasks where adding a chain-of-thought might hurt performance compared to standard prompting (Ye & Durrett, 2022). We also show self-consistency significantly outperforms sample-and-rank, beam search, ensemble-based approaches, and is robust to sampling strategies and imperfect prompts.

## 2 SELF-CONSISTENCY OVER DIVERSE REASONING PATHS

A salient aspect of humanity is that people think differently. It is natural to suppose that in tasks requiring deliberate thinking, there are likely several ways to attack the problem. We propose that such a process can be simulated in language models via sampling from the language model's decoder. For instance, as shown in Figure 1, a model can generate several plausible responses to a math question that all arrive at the same correct answer (Outputs 1 and 3). Since language models are not perfect reasoners, the model might also produce an incorrect reasoning path or make a mistake in one of the reasoning steps (e.g., in Output 2), but such solutions are less likely to arrive at the *same* answer. That is, we hypothesize that correct reasoning processes, even if they are diverse, tend to have greater agreement in their final answer than incorrect processes.

We leverage this intuition by proposing the following *self-consistency* method. First, a language model is prompted with a set of manually written chain-of-thought exemplars (Wei et al., 2022). Next,

|  | GSM8K | MultiArith | AQuA | SVAMP | CSQA | ARC-c |
|---|---|---|---|---|---|---|
| Greedy decode | 56.5 | 94.7 | 35.8 | 79.0 | 79.0 | 85.2 |
| Weighted avg (unnormalized) | $56.3 \pm 0.0$ | $90.5 \pm 0.0$ | $35.8 \pm 0.0$ | $73.0 \pm 0.0$ | $74.8 \pm 0.0$ | $82.3 \pm 0.0$ |
| Weighted avg (normalized) | $22.1 \pm 0.0$ | $59.7 \pm 0.0$ | $15.7 \pm 0.0$ | $40.5 \pm 0.0$ | $52.1 \pm 0.0$ | $51.7 \pm 0.0$ |
| Weighted sum (unnormalized) | $59.9 \pm 0.0$ | $92.2 \pm 0.0$ | $38.2 \pm 0.0$ | $76.2 \pm 0.0$ | $76.2 \pm 0.0$ | $83.5 \pm 0.0$ |
| Weighted sum (normalized) | $74.1 \pm 0.0$ | $99.3 \pm 0.0$ | $48.0 \pm 0.0$ | $86.8 \pm 0.0$ | $80.7 \pm 0.0$ | $88.7 \pm 0.0$ |
| Unweighted sum (majority vote) | $74.4 \pm 0.1$ | $99.3 \pm 0.0$ | $48.3 \pm 0.5$ | $86.6 \pm 0.1$ | $80.7 \pm 0.1$ | $88.7 \pm 0.1$ |

Table 1: Accuracy comparison of different answer aggregation strategies on PaLM-540B.

we sample a set of candidate outputs from the language model's decoder, generating a diverse set of candidate reasoning paths. Self-consistency is compatible with most existing sampling algorithms, including temperature sampling (Ackley et al., 1985; Ficler & Goldberg, 2017), top-$k$ sampling (Fan et al., 2018; Holtzman et al., 2018; Radford et al., 2019), and nucleus sampling (Holtzman et al., 2020). Finally, we aggregate the answers by marginalizing out the sampled reasoning paths and choosing the answer that is the most consistent among the generated answers.

In more detail, assume the generated answers $\mathbf{a}_i$ are from a fixed answer set, $\mathbf{a}_i \in \mathbb{A}$, where $i = 1, \ldots, m$ indexes the $m$ candidate outputs sampled from the decoder. Given a prompt and a question, self-consistency introduces an additional latent variable $\mathbf{r}_i$, which is a sequence of tokens representing the reasoning path in the $i$-th output, then couples the generation of $(\mathbf{r}_i, \mathbf{a}_i)$ where $\mathbf{r}_i \to \mathbf{a}_i$, i.e., generating a reasoning path $\mathbf{r}_i$ is optional and only used to reach the final answer $\mathbf{a}_i$. As an example, consider Output 3 from Figure 1: the first few sentences "*She eats 3 for breakfast ... So she has 9 eggs * \$2 = \$18.*" constitutes $\mathbf{r}_i$, while the answer *18* from the last sentence, "*The answer is \$18*", is parsed as $\mathbf{a}_i$.[1] After sampling multiple $(\mathbf{r}_i, \mathbf{a}_i)$ from the model's decoder, self-consistency applies a marginalization over $\mathbf{r}_i$ by taking a majority vote over $\mathbf{a}_i$, i.e., $\arg\max_a \sum_{i=1}^m \mathbb{1}(\mathbf{a}_i = a)$, or as we defined as the most "consistent" answer among the final answer set.

In Table 1, we show the test accuracy over a set of reasoning tasks by using different answer aggregation strategies. In addition to majority vote, one can also weight each $(\mathbf{r}_i, \mathbf{a}_i)$ by $P(\mathbf{r}_i, \mathbf{a}_i \mid \text{prompt}, \text{question})$ when aggregating the answers. Note to compute $P(\mathbf{r}_i, \mathbf{a}_i \mid \text{prompt}, \text{question})$, we can either take the unnormalized probability of the model generating $(\mathbf{r}_i, \mathbf{a}_i)$ given (prompt, question), or we can normalize the conditional probability by the output length (Brown et al., 2020), i.e.,

$$P(\mathbf{r}_i, \mathbf{a}_i \mid \text{prompt}, \text{question}) = \exp^{\frac{1}{K} \sum_{k=1}^K \log P(t_k \mid \text{prompt}, \text{question}, t_1, \ldots, t_{k-1})}, \tag{1}$$

where $\log P(t_k \mid \text{prompt}, \text{question}, t_1, \ldots, t_{k-1})$ is the log probability of generating the $k$-th token $t_k$ in $(\mathbf{r}_i, \mathbf{a}_i)$ conditioned on the previous tokens, and $K$ is the total number of tokens in $(\mathbf{r}_i, \mathbf{a}_i)$. In Table 1, we show that taking the "unweighted sum", i.e., taking a majority vote directly over $\mathbf{a}_i$ yields a very similar accuracy as aggregating using the "normalized weighted sum". We took a closer look at the model's output probabilities and found this is because for each $(\mathbf{r}_i, \mathbf{a}_i)$, the normalized conditional probabilities $P(\mathbf{r}_i, \mathbf{a}_i \mid \text{prompt}, \text{question})$ are quite close to each other, i.e., the language model regards those generations as "similarly likely".[2] Additionally, when aggregating the answers, the results in Table 1 show that the "normalized" weighted sum (i.e., Equation 1) yields a much higher accuracy compared to its unnormalized counterpart. For completeness, in Table 1 we also report the results by taking a "weighted average", i.e., each $a$ gets a score of its weighted sum divided by $\sum_{i=1}^m \mathbb{1}(\mathbf{a}_i = a)$, which results in a much worse performance.

Self-consistency explores an interesting space between open-ended text generation and optimal text generation with a fixed answer. Reasoning tasks typically have fixed answers, which is why researchers have generally considered greedy decoding approaches (Radford et al., 2019; Wei et al., 2022; Chowdhery et al., 2022). However, we have found that even when the desired answer is fixed, introducing diversity in the reasoning processes can be highly beneficial; therefore we leverage

---

[1]The parser is task dependent. For arithmetic reasoning, we parse the first numerical part as the final answer after the model generates "The answer is ". For commonsense reasoning, we parse the full string answer as the final answer after the model generates "The answer is ". Most generated outputs have a consistent format of "{Reasoning paths}. The answer is X." if we prompt the language model in this format.

[2]This also means that the language model is not well calibrated and thus cannot distinguish well between correct solutions and wrong solutions, which also explains why additional re-rankers were trained to better judge the quality of the solutions in previous work (Cobbe et al., 2021; Thoppilan et al., 2022).

sampling, as commonly used for open-ended text generation (Radford et al., 2019; Brown et al., 2020; Thoppilan et al., 2022), to achieve this goal. One should note that self-consistency can be applied only to problems where the final answer is from a fixed answer set, but in principle this approach can be extended to open-text generation problems if a good metric of consistency can be defined between multiple generations, e.g., whether two answers agree or contradict each other.

## 3  EXPERIMENTS

We conducted a series of experiments to compare the proposed self-consistency method with existing approaches on a range of reasoning benchmarks. We find that self-consistency robustly improves reasoning accuracy for every language model considered, spanning a wide range of model scales.

### 3.1  EXPERIMENT SETUP

**Tasks and datasets.**  We evaluate self-consistency on the following reasoning benchmarks.[3]

- **Arithmetic reasoning**. For these tasks, we used the Math Word Problem Repository (Koncel-Kedziorski et al., 2016), including AddSub (Hosseini et al., 2014), MultiArith (Roy & Roth, 2015), and ASDiv (Miao et al., 2020). We also included AQUA-RAT (Ling et al., 2017), a recently published benchmark of grade-school-math problems (GSM8K; Cobbe et al., 2021), and a challenge dataset over math word problems (SVAMP; Patel et al., 2021).
- **Commonsense reasoning**. For these tasks, we used CommonsenseQA (Talmor et al., 2019), StrategyQA (Geva et al., 2021), and the AI2 Reasoning Challenge (ARC) (Clark et al., 2018).
- **Symbolic Reasoning**. We evaluate two symbolic reasoning tasks: last letter concatenation (e.g., the input is "Elon Musk" and the output should be "nk"), and Coinflip (e.g., a coin is heads-up, after a few flips is the coin still heads-up?) from Wei et al. (2022).

**Language models and prompts.**  We evaluate self-consistency over four transformer-based language models with varying scales:

- UL2 (Tay et al., 2022) is an encoder-decoder model trained on a mixture of denoisers with 20-billion parameters. UL2 is completely open-sourced[4] and has similar or better performance than GPT-3 on zero-shot SuperGLUE, with only 20B parameters and thus is more compute-friendly;
- GPT-3 (Brown et al., 2020) with 175-billion parameters. We use two public engines *code-davinci-001* and *code-davinci-002* from the Codex series (Chen et al., 2021) to aid reproducibility;[5]
- LaMDA-137B (Thoppilan et al., 2022) is a dense left-to-right, decoder-only language model with 137-billion parameters, pre-trained on a mixture of web documents, dialog data and Wikipedia;
- PaLM-540B (Chowdhery et al., 2022) is a dense left-to-right, decoder-only language model with 540-billion parameters, pre-trained on a high quality corpus of 780 billion tokens with filtered webpages, books, Wikipedia, news articles, source code, and social media conversations.

We perform all experiments in the few-shot setting, without training or fine-tuning the language models. For a fair comparison we use the same prompts as in Wei et al. (2022): for all arithmetic reasoning tasks we use the same set of 8 manually written exemplars; for each commonsense reasoning task, 4-7 exemplars are randomly chosen from the training set with manually composed chain-of-thought prompts.[6] Full details on the prompts used are given in Appendix A.3.

**Sampling scheme.**  To sample diverse reasoning paths, we followed similar settings to those suggested in Radford et al. (2019); Holtzman et al. (2020) for open-text generation. In particular, for UL2-20B and LaMDA-137B we applied temperature sampling with $T = 0.5$ and truncated at the top-$k$ ($k = 40$) tokens with the highest probability, for PaLM-540B we applied $T = 0.7, k = 40$, and for GPT-3 we use $T = 0.7$ without top-$k$ truncation. We provide an ablation study in Section 3.5 to show that self-consistency is generally robust to sampling strategies and parameters.

---

[3]By default we use the test split for all datasets if the labels are available for evaluation. For CommonsenseQA we use the dev split; for StrategyQA we use the question-only set from BIG-bench collaboration (2021): https://github.com/google/BIG-bench/tree/main/bigbench/benchmark_tasks/strategyqa.

[4]Model checkpoints at https://github.com/google-research/google-research/tree/master/ul2.

[5]Public API available at https://openai.com/api/.

[6]Self-consistency is robust to different sets of prompts and we provide a study in Appendix A.1.2.

## 3.2 MAIN RESULTS

We report the results of self-consistency averaged over 10 runs, where we sampled 40 outputs independently from the decoder in each run. The baseline we compare to is chain-of-thought prompting with greedy decoding (Wei et al., 2022), referred to as **CoT-prompting**, which has been previously used for decoding in large language models (Chowdhery et al., 2022).

**Arithmetic Reasoning** The results are shown in Table 2.[7] Self-consistency improves the arithmetic reasoning performance over **all four language models** significantly over chain-of-thought prompting. More surprisingly, the gains become more significant when the language model's scale increases, e.g., we see +3%-6% absolute accuracy improvement over UL2-20B but +9%-23% for LaMDA-137B and GPT-3. For larger models that already achieve high accuracy on most tasks (e.g., GPT-3 and PaLM-540B), self-consistency still contributes significant additional gains with +12%-18% absolute accuracy on tasks like AQuA and GSM8K, and +7%-11% on SVAMP and ASDiv. With self-consistency, we achieve new state-of-the-art results on almost all tasks: despite the fact that self-consistency is unsupervised and task-agnostic, these results compare favorably to existing approaches that require task-specific training, or fine-tuning with thousands of examples (e.g., on GSM8K).

| | Method | AddSub | MultiArith | ASDiv | AQuA | SVAMP | GSM8K |
|---|---|---|---|---|---|---|---|
| | Previous SoTA | **94.9**[a] | 60.5[a] | 75.3[b] | 37.9[c] | 57.4[d] | 35[e] / 55[g] |
| UL2-20B | CoT-prompting | 18.2 | 10.7 | 16.9 | 23.6 | 12.6 | 4.1 |
| | Self-consistency | 24.8 (+6.6) | 15.0 (+4.3) | 21.5 (+4.6) | 26.9 (+3.3) | 19.4 (+6.8) | 7.3 (+3.2) |
| LaMDA-137B | CoT-prompting | 52.9 | 51.8 | 49.0 | 17.7 | 38.9 | 17.1 |
| | Self-consistency | 63.5 (+10.6) | 75.7 (+23.9) | 58.2 (+9.2) | 26.8 (+9.1) | 53.3 (+14.4) | 27.7 (+10.6) |
| PaLM-540B | CoT-prompting | 91.9 | 94.7 | 74.0 | 35.8 | 79.0 | 56.5 |
| | Self-consistency | 93.7 (+1.8) | 99.3 (+4.6) | 81.9 (+7.9) | 48.3 (+12.5) | 86.6 (+7.6) | 74.4 (+17.9) |
| GPT-3 Code-davinci-001 | CoT-prompting | 57.2 | 59.5 | 52.7 | 18.9 | 39.8 | 14.6 |
| | Self-consistency | 67.8 (+10.6) | 82.7 (+23.2) | 61.9 (+9.2) | 25.6 (+6.7) | 54.5 (+14.7) | 23.4 (+8.8) |
| GPT-3 Code-davinci-002 | CoT-prompting | 89.4 | 96.2 | 80.1 | 39.8 | 75.8 | 60.1 |
| | Self-consistency | 91.6 (+2.2) | **100.0** (+3.8) | **87.8** (+7.6) | **52.0** (+12.2) | **86.8** (+11.0) | **78.0** (+17.9) |

Table 2: Arithmetic reasoning accuracy by self-consistency compared to chain-of-thought prompting (Wei et al., 2022). The previous SoTA baselines are obtained from: $a$: Relevance and LCA operation classifier (Roy & Roth, 2015), $b$: Lan et al. (2021), $c$: Amini et al. (2019), $d$: Pi et al. (2022), $e$: GPT-3 175B finetuned with 7.5k examples (Cobbe et al., 2021), $g$: GPT-3 175B finetuned plus an additional 175B verifier (Cobbe et al., 2021). The best performance for each task is shown in bold.

| | Method | CSQA | StrategyQA | ARC-e | ARC-c | Letter (4) | Coinflip (4) |
|---|---|---|---|---|---|---|---|
| | Previous SoTA | **91.2**[a] | 73.9[b] | 86.4[c] | 75.0[c] | N/A | N/A |
| UL2-20B | CoT-prompting | 51.4 | 53.3 | 61.6 | 42.9 | 0.0 | 50.4 |
| | Self-consistency | 55.7 (+4.3) | 54.9 (+1.6) | 69.8 (+8.2) | 49.5 (+6.8) | 0.0 (+0.0) | 50.5 (+0.1) |
| LaMDA-137B | CoT-prompting | 57.9 | 65.4 | 75.3 | 55.1 | 8.2 | 72.4 |
| | Self-consistency | 63.1 (+5.2) | 67.8 (+2.4) | 79.3 (+4.0) | 59.8 (+4.7) | 8.2 (+0.0) | 73.5 (+1.1) |
| PaLM-540B | CoT-prompting | 79.0 | 75.3 | 95.3 | 85.2 | 65.8 | 88.2 |
| | Self-consistency | 80.7 (+1.7) | **81.6** (+6.3) | **96.4** (+1.1) | **88.7** (+3.5) | 70.8 (+5.0) | 91.2 (+3.0) |
| GPT-3 Code-davinci-001 | CoT-prompting | 46.6 | 56.7 | 63.1 | 43.1 | 7.8 | 71.4 |
| | Self-consistency | 54.9 (+8.3) | 61.7 (+5.0) | 72.1 (+9.0) | 53.7 (+10.6) | 10.0 (+2.2) | 75.9 (+4.5) |
| GPT-3 Code-davinci-002 | CoT-prompting | 79.0 | 73.4 | 94.0 | 83.6 | 70.4 | 99.0 |
| | Self-consistency | 81.5 (+2.5) | 79.8 (+6.4) | 96.0 (+2.0) | 87.5 (+3.9) | **73.4** (+3.0) | **99.5** (+0.5) |

Table 3: Commonsense and symbolic reasoning accuracy by self-consistency compared to chain-of-thought prompting (Wei et al., 2022). The previous SoTA baselines are obtained from: $a$: DeBERTaV3-large + KEAR (Xu et al., 2021b), $b$: Chowdhery et al. (2022), $c$: UnifiedQA-FT (Khashabi et al., 2020). The best performance for each task is shown in bold.

---

[7]The standard deviation of self-consistency is $\leq 0.5$ for all tasks and is thus omitted in the table. Please refer to Figure 2, Figure 7 and 8 for the standard deviations under varying numbers of sampled paths.

**Commonsense and Symbolic Reasoning**    Table 3 shows the results on commonsense and symbolic reasoning tasks. Similarly, self-consistency yields large gains across all four language models, and obtained SoTA results on 5 out of 6 tasks. For symbolic reasoning, we test the out-of-distribution (OOD) setting where the input prompt contains examples of 2-letters or 2-flips but we test examples of 4-letters and 4-flips (this setting is more challenging as PaLM-540B or GPT-3 can already achieve perfect in-distribution accuracy). In this challenging OOD setting, the gain of self-consistency is still quite significant compared to CoT-prompting with sufficient model sizes.

To show the effect of the number of sampled reasoning paths, we plot the accuracy (mean and standard deviation over 10 runs) with respect to varying numbers of sampled paths (1, 5, 10, 20, 40) in Figure 2. The results show that sampling a higher number (e.g., 40) of reasoning paths leads to a consistently better performance, further emphasizing the importance of introducing diversity in the reasoning paths. In Table 4, we show self-consistency yields a richer set of reasoning paths compared to greedy decoding with a few example questions from two tasks.

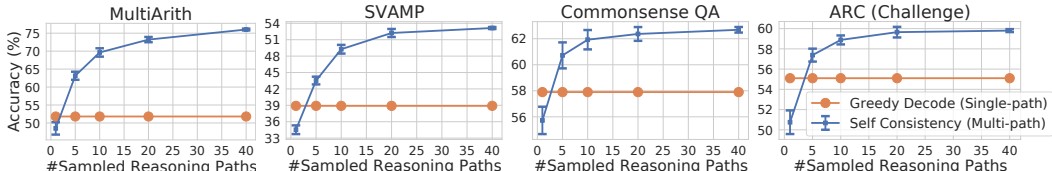

Figure 2: Self-consistency (blue) significantly improves accuracy over CoT-prompting with greedy decoding (orange) across arithmetic and commonsense reasoning tasks, over LaMDA-137B. Sampling a higher number of diverse reasoning paths consistently improves reasoning accuracy.

|  | GSM8K | StrategyQA |
|---|---|---|
| **Question** | *Henry made two stops during his 60-mile bike trip. He first stopped after 20 miles. His second stop was 15 miles before the end of the trip. How many miles did he travel between his first and second stops?* | *Yes or no: Is Albany, Georgia the most populous US Albany?* |
| **Greedy Decode** | He traveled 60 miles in total. He stopped after 20 miles, so he traveled 60 - 20 = 40 miles between the first and second stops. The answer is 40. | Albany, Georgia is the most populous US Albany. Thus, the answer is yes. |
| **Sampled Path 1** | Henry travelled a total of 60 miles. His first stop was 20 miles in, and his second stop was 15 miles before the end. So between his first and second stops he travelled 60 - 20 -15 = 25 miles. The answer is 25. | The most populous US Albany is Albany, New York. Thus, Albany, Georgia is not the most populous US Albany. So the answer is no. |
| **Sampled Path 2** | He made two stops during a 60-mile trip. The first was 20 miles into the trip. The second was 15 miles before the end of the trip. This means the second stop was 60 - 15 = 45 miles into the trip. Since he made the stops in order, the second stop must have been 45 - 20 = 25 miles after the first stop. The answer is 25. | Albany, Georgia has a population of about 88,000. Albany, New York has a population of about 95,000. Thus, Albany, Georgia is not the most populous US Albany. So the answer is no. |

Table 4: Examples where self-consistency helps repair the errors over greedy decode, on PaLM-540B. Two sampled reasoning paths that are consistent with the ground truth are shown.

## 3.3    SELF-CONSISTENCY HELPS WHEN CHAIN-OF-THOUGHT HURTS PERFORMANCE

Ye & Durrett (2022) show that sometimes chain-of-thought prompting could hurt performance compared to standard prompting in few-shot in-context learning. Here we perform a study using self-consistency to see if it can help fill in the gap, over a set of common NLP tasks, including (1) Closed-Book Question Answering: BoolQ (Clark et al., 2019), HotpotQA (Yang et al., 2018), and (2) Natural Language Inference: e-SNLI (Camburu et al., 2018), ANLI (Nie et al., 2020) and RTE (Dagan et al., 2005; Bar-Haim et al., 2006; Giampiccolo et al., 2007; Bentivogli et al., 2009).

The results over PaLM-540B are shown in Table 5. For some tasks (e.g., ANLI-R1, e-SNLI, RTE), adding chain-of-thought does hurt performance compared to standard prompting (Brown et al., 2020), but self-consistency is able to robustly boost the performance and outperform standard prompting, making it a reliable way to add rationales in few-shot in-context learning for common NLP tasks.

|  | ANLI R1 / R2 / R3 | e-SNLI | RTE | BoolQ | HotpotQA (EM/F1) |
|---|---|---|---|---|---|
| Standard-prompting (no-rationale) | 69.1 / 55.8 / 55.8 | 85.8 | 84.8 | 71.3 | 27.1 / 36.8 |
| CoT-prompting (Wei et al., 2022) | 68.8 / 58.9 / 60.6 | 81.0 | 79.1 | 74.2 | 28.9 / 39.8 |
| Self-consistency | **78.5 / 64.5 / 63.4** | **88.4** | **86.3** | **78.4** | **33.8 / 44.6** |

Table 5: Compare Standard/CoT prompting with self-consistency on common NLP tasks.

## 3.4 COMPARE TO OTHER EXISTING APPROACHES

We conduct a set of additional studies and show that self-consistency significantly outperforms existing methods including sample-and-rank, beam search, and ensemble-based approaches.

**Comparison to Sample-and-Rank** One commonly used approach to improve generation quality is sample-and-rank, where multiple sequences are sampled from the decoder and then ranked according to each sequence's log probability (Adiwardana et al., 2020). We compare self-consistency with sample-and-rank on GPT-3 *code-davinci-001*, by sampling the same number of sequences from the decoder as self-consistency and taking the final answer from the top-ranked sequence. The results are shown in Figure 3. While sample-and-rank does improve the accuracy with additionally sampled sequences and ranking, the gain is much smaller compared to self-consistency.

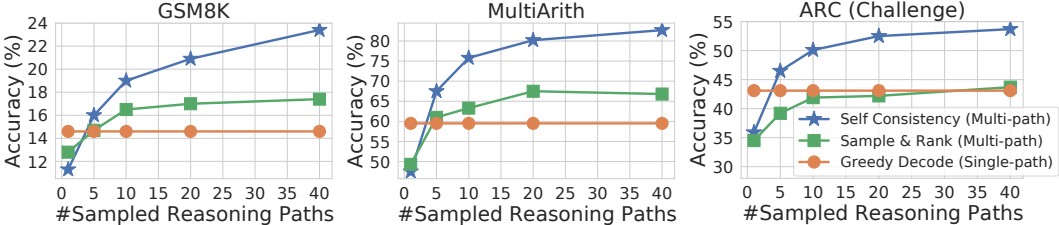

Figure 3: Self-consistency significantly outperforms sample-and-rank with the same # of samples.

**Comparison to Beam Search** In Table 6, we compare self-consistency with beam search decoding on the UL2-20B model. For a fair comparison we report the accuracy under the same number of beams and reasoning paths. On both tasks self-consistency outperforms beam search significantly. Note self-consistency can also adopt beam search to decode each reasoning path (results are shown as "Self-consistency using beam search"), but its performance is worse compared to self-consistency with sampling. The reason is that beam search yields a lower diversity in the outputs (Li & Jurafsky, 2016), while in self-consistency the diversity of the reasoning paths is the key to a better performance.

| | Beam size / Self-consistency paths | 1 | 5 | 10 | 20 | 40 |
|---|---|---|---|---|---|---|
| | Beam search decoding (top beam) | 23.6 | 19.3 | 16.1 | 15.0 | 10.2 |
| AQuA | Self-consistency using beam search | 23.6 | 19.8 ± 0.3 | 21.2 ± 0.7 | 24.6 ± 0.4 | 24.2 ± 0.5 |
| | Self-consistency using sampling | 19.7 ± 2.5 | **24.9 ± 2.6** | **25.3 ± 1.8** | **26.7 ± 1.0** | **26.9 ± 0.5** |
| | Beam search decoding (top beam) | 10.7 | 12.0 | 11.3 | 11.0 | 10.5 |
| MultiArith | Self-consistency using beam search | 10.7 | 11.8 ± 0.0 | 11.4 ± 0.1 | 12.3 ± 0.1 | 10.8 ± 0.1 |
| | Self-consistency using sampling | 9.5 ± 1.2 | 11.3 ± 1.2 | **12.3 ± 0.8** | **13.7 ± 0.9** | **14.7 ± 0.3** |

Table 6: Compare self-consistency with beam search decoding on the UL2-20B model.

**Comparison to Ensemble-based Approaches** We further compare self-consistency to ensemble-based methods for few-shot learning. In particular, we consider ensembling by: (1) prompt order permutation: we randomly permute the exemplars in the prompt 40 times to mitigate model's sensitivity to prompt order (Zhao et al., 2021; Lu et al., 2021); and (2) multiple sets of prompts (Gao et al., 2021): we manually write 3 different sets of prompts. We took majority vote of the answers from greedy decoding in both approaches as an ensemble. Table 7 shows that compared to self-consistency, existing ensemble-based approaches achieve a much smaller gain.[8] In addition, note that self-consistency is different from a typical model-ensemble approach, where *multiple* models are trained and their outputs are aggregated. Self-consistency acts more like a "self-ensemble" on top of a *single* language model. We additionally show the results of ensembling multiple models in Appendix A.1.3 where the model-ensembles perform much worse compared to self-consistency.

| | GSM8K | MultiArith | SVAMP | ARC-e | ARC-c |
|---|---|---|---|---|---|
| CoT (Wei et al., 2022) | 17.1 | 51.8 | 38.9 | 75.3 | 55.1 |
| Ensemble (3 sets of prompts) | 18.6 ± 0.5 | 57.1 ± 0.7 | 42.1 ± 0.6 | 76.6 ± 0.1 | 57.0 ± 0.2 |
| Ensemble (40 prompt permutations) | 19.2 ± 0.1 | 60.9 ± 0.2 | 42.7 ± 0.1 | 76.9 ± 0.1 | 57.0 ± 0.1 |
| Self-Consistency (40 sampled paths) | **27.7 ± 0.2** | **75.7 ± 0.3** | **53.3 ± 0.2** | **79.3 ± 0.3** | **59.8 ± 0.2** |

Table 7: Self-consistency outperforms prompt-order and multi-prompt ensembles on LaMDA-137B.

---

[8]Self-consistency is compatible with both ensemble approaches and we show the results in Appendix A.1.4.

## 3.5 ADDITIONAL STUDIES

We conducted a number of additional experiments to analyze different aspects of the self-consistency method, including its robustness to sampling strategies and parameters, and how it works with imperfect prompts and non-natural-language reasoning paths.

**Self-Consistency is Robust to Sampling Strategies and Scaling**   We show self-consistency is robust to sampling strategies and parameters, by varying $T$ in temperature sampling (Ackley et al., 1985; Ficler & Goldberg, 2017), $k$ in top-$k$ sampling (Fan et al., 2018; Holtzman et al., 2018; Radford et al., 2019), and $p$ in nucleus sampling (Holtzman et al., 2020), over PaLM-540B in Figure 4 (left). Figure 4 (right) shows that self-consistency robustly improves performance across all scales for the LaMDA-137B model series. The gain is relatively lower for smaller models due to certain abilities (e.g., arithmetic) only emerge when the model reaches a sufficient scale (Brown et al., 2020).

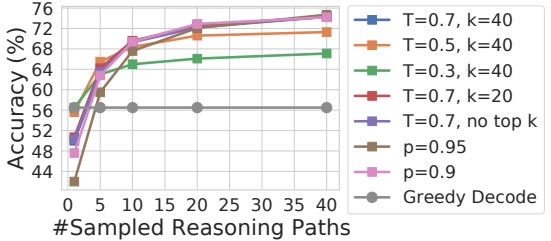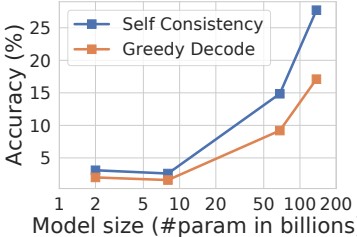

Figure 4: GSM8K accuracy. (Left) Self-consistency is robust to various sampling strategies and parameters. (Right) Self-consistency improves performance across language model scales.

**Self-Consistency Improves Robustness to Imperfect Prompts**   For few-shot learning with manually constructed prompts, human annotators sometimes make minor mistakes when creating the prompts. We further study if self-consistency can help improve a language model's robustness to imperfect prompts.[9] We show the results in Table 8: while imperfect prompts decrease accuracy with greedy decoding ($17.1 \rightarrow 14.9$), self-consistency can fill in the gaps and robustly improve the results.

Additionally, we found that the consistency (in terms of % of decodes agreeing with the final aggregated answer) is highly correlated with accuracy (Figure 5, over GSM8K). This suggests that one can use self-consistency to provide an *uncertainty estimate* of the model in its generated solutions. In other words, one can use low consistency as an indicator that the model has low confidence; i.e., self-consistency confers some ability for the model to "know when it doesn't know".

| | | |
|---|---|---|
| LaMDA-137B | Prompt with correct chain-of-thought | 17.1 |
| | Prompt with imperfect chain-of-thought | 14.9 |
| | + Self-consistency (40 paths) | **23.4** |
| | Prompt with equations | 5.0 |
| | + Self-consistency (40 paths) | **6.5** |
| PaLM-540B | Zero-shot CoT (Kojima et al., 2022) | 43.0 |
| | + Self-consistency (40 paths) | **69.2** |

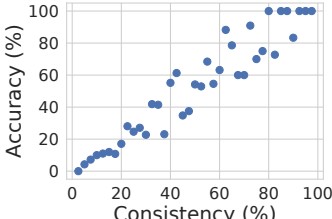

Table 8: Self-consistency works under imperfect prompts, equation prompts and zero-shot chain-of-thought for GSM8K.

Figure 5: The consistency is correlated with model's accuracy.

**Self-Consistency Works for Non-Natural-Language Reasoning Paths and Zero-shot CoT**   We also tested the generality of the self-consistency concept to alternative forms of intermediate reasoning like equations (e.g., from "*There are 3 cars in the parking lot already. 2 more arrive. Now there are 3 + 2 = 5 cars.*" to "*3 + 2 = 5*"). The results are shown in Table 8 ("Prompt with equations"): self-consistency still improves accuracy by generating intermediate equations; however, compared to generating natural language reasoning paths, the gain is smaller since the equations are much shorter and less opportunity remains for generating diversity in the decoding process. In addition, we tested self-consistency with zero-shot chain-of-thought (Kojima et al., 2022) and show that self-consistency works for zero-shot CoT as well and improves the results significantly (+26.2%) in Table 8.

---

[9]We use the same prompts as before, but swap all the numbers in the reasoning paths with random numbers except the final answer, e.g., from "*There are 3 cars in the parking lot already. 2 more arrive. Now there are 3 + 2 = 5 cars.*" to "*There are 7 cars in the parking lot already. 6 more arrive. Now there are 7 + 6 = 5 cars.*".

## 4 RELATED WORK

**Reasoning in language models.** Language models are known to struggle in Type 2 tasks, such as arithmetic, logical and commonsense reasoning (Evans, 2010). Previous work has primarily focused on *specialized* approaches for improving reasoning (Andor et al., 2019; Ran et al., 2019; Geva et al., 2020; Piękos et al., 2021). Compared to prior work, self-consistency is applicable to a wide range of reasoning tasks without any additional supervision or fine-tuning, while still substantially improving the performance of the chain-of-thought prompting approach proposed in Wei et al. (2022).

**Sampling and re-ranking in language models.** Multiple decoding strategies for language models have been proposed in the literature, e.g., temperature sampling (Ackley et al., 1985; Ficler & Goldberg, 2017), top-$k$ sampling (Fan et al., 2018; Holtzman et al., 2018; Radford et al., 2019), nucleus sampling (Holtzman et al., 2020), minimum Bayes risk decoding (Eikema & Aziz, 2020; Shi et al., 2022), and typical decoding (Meister et al., 2022). Other work has sought to explicitly promote diversity in the decoding process (Batra et al., 2012; Li et al., 2016; Vijayakumar et al., 2018).

Re-ranking is another common approach to improve generation quality in language models (Adiwardana et al., 2020; Shen et al., 2021). Thoppilan et al. (2022) collect additional human annotations to train a re-ranker for response filtering. Cobbe et al. (2021) train a "verifier" to re-rank generated solutions, which substantially improves the solve rate on math tasks compared to just fine-tuning the language model. Elazar et al. (2021) improve the consistency of factual knowledge extraction by extending pre-training with an additional consistency loss. All these methods require either training an additional re-ranker or collecting additional human annotation, while self-consistency requires no additional training, fine-tuning, nor extra data collection.

**Extract reasoning paths.** Some previous work has considered task-specific approaches for identifying reasoning paths, such as constructing semantic graphs (Xu et al., 2021a), learning an RNN to retrieve reasoning paths over the Wikipedia graph (Asai et al., 2020), fine-tuning with human annotated reasoning paths on math problems (Cobbe et al., 2021), or training an extractor with heuristic-based pseudo reasoning paths (Chen et al., 2019). More recently, the importance of diversity in the reasoning processes has been noticed, but only leveraged via task-specific training, either through an additional QA model over extracted reasoning paths (Chen et al., 2019), or by the introduction of latent variables in a commonsense knowledge graph (Yu et al., 2022). Compared to these approaches, self-consistency is far simpler and requires no additional training. The approach we propose simply couples the generation of reasoning paths and a final answer by sampling from the decoder, using aggregation to recover the most consistent answer without additional modules.

**Consistency in language models.** Some prior work has shown that language models can suffer from inconsistency in conversation (Adiwardana et al., 2020), explanation generation (Camburu et al., 2020), and factual knowledge extraction (Elazar et al., 2021). Welleck et al. (2020) use "consistency" to refer to generating an infinite-length sequence in recurrent language models. Nye et al. (2021) improve the logical consistency of samples from a System 1 model by adding a System 2-inspired logical reasoning module. In this paper we focus on a slightly different notion of "consistency", i.e., utilizing answer consistency among diverse reasoning paths to improve accuracy.

## 5 CONCLUSION AND DISCUSSION

We introduced a simple yet effective method called self-consistency, and observed that it significantly improves accuracy in a range of arithmetic and commonsense reasoning tasks, across four large language models with varying scales. Beyond accuracy gains, self-consistency is also useful for collecting rationales when performing reasoning tasks with language models, and for providing uncertainty estimates and improved calibration of language model outputs.

One limitation of self-consistency is that it incurs more computation cost. In practice people can try a small number of paths (e.g., 5 or 10) as a starting point to realize most of the gains while not incurring too much cost, as in most cases the performance saturates quickly (Figure 2). As part of future work, one could use self-consistency to generate better supervised data to fine-tune the model, such that the model can give more accurate predictions in a single inference run after fine-tuning. In addition, we observed that language models can sometimes generate incorrect or nonsensical reasoning paths (e.g., the StrategyQA example in Table 4, the two population numbers are not exactly correct), and further work is needed to better ground models' rationale generations.

## REPRODUCIBILITY STATEMENT

In experiments, we included four different language models with varying scales. Two of them are public models: UL2 is a completely open-sourced model with model checkpoints available at `https://github.com/google-research/google-research/tree/master/ul2`; GPT-3 is also a public model with public API available at `https://openai.com/api/`. For GPT-3, we have included two public engines ("code-davinci-001" and "code-davinci-002") to further aid reproducibility, as Codex is currently free so anyone can reproduce the results. In addition, as our results make use of LaMDA-137B and PaLM-540B that are not publicly available, we provide the exact input prompts for all tasks in Appendix A.3 (and note that we do not perform any finetuning and only apply prompting to off-the-shelf language models).

## ETHICS STATEMENT

As we stated in the discussion, language models can sometimes generate nonsensical or non-factual reasoning paths, so one should use language models' outputs with extra caution. We deal with reasoning tasks mostly and the generated rationales are only used for inspecting how a model reaches its answer. One could potentially use the generated rationales to further check why the model makes certain mistakes or whether the model contains any biases when performing a certain task. For language model in real-world use, further work is needed to better ground models' predictions and improve model's factuality and safety, to ensure the models do not cause harms to users.

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

## A APPENDIX

### A.1 ADDITIONAL EXPERIMENT RESULTS

#### A.1.1 ROBUSTNESS TO SAMPLING STRATEGIES AND PARAMETERS

In Figure 6 we ablate the results with respect to different sampling strategies and parameters by varying $T$ in temperature sampling and $k$ in Top-$k$ sampling, on LaMDA-137B. We show that self-consistency is robust to various sampling strategies and parameters.

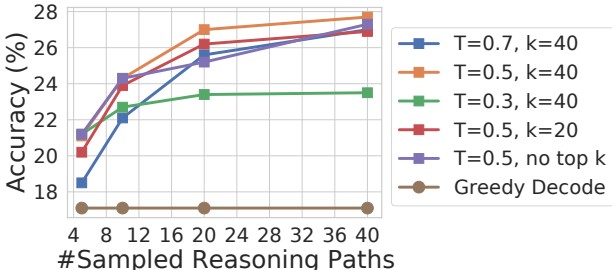

Figure 6: GSM8K accuracy over LaMDA-137B. Self-consistency works under various sampling strategies and sampling parameters.

In Figure 7 and Figure 8, we show the results of self-consistency compared with greedy decoding a single path over LaMDA-137B and PaLM-540B, respectively. Self-consistency improves over greedy decode by a quite significant margin on both models, on top of high accuracy already achieved by scaling up model sizes.

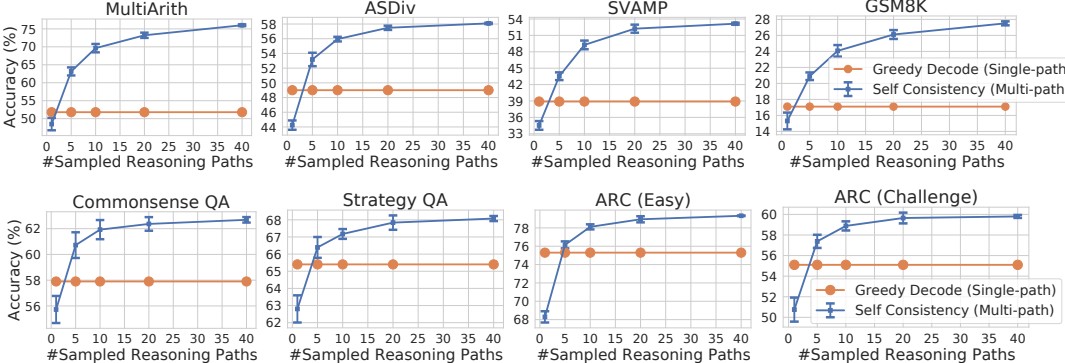

Figure 7: Self-consistency (blue) significantly improves accuracy across various arithmetic and commonsense reasoning tasks, over LaMDA-137B. Sampling a higher number of diverse reasoning paths consistently improves reasoning accuracy.

We further show additional sampled reasoning paths from the LaMDA-137B model in Table 12, and sampled reasoning paths from the PaLM-540B model in Table 13. We see that the diversity in the additionally sampled reasoning paths indeed helps the model arrive at a more correct final answer after aggregation.

#### A.1.2 ROBUSTNESS TO DIFFERENT SETS OF PROMPTS

In Table 9, we further show that self-consistency is quite robust to different sets of input prompts. We manually wrote 3 different sets of chain-of-thought as prompts to the model. Across all sets of prompts, self-consistency yields consistent gains over the original CoT approach.

#### A.1.3 COMPARED TO MODEL ENSEMBLES

Additionally, we provide results of directly ensembling the outputs from **multiple language models**. The results are shown in Table 10, by greedily decoding sequences from 3 language models and

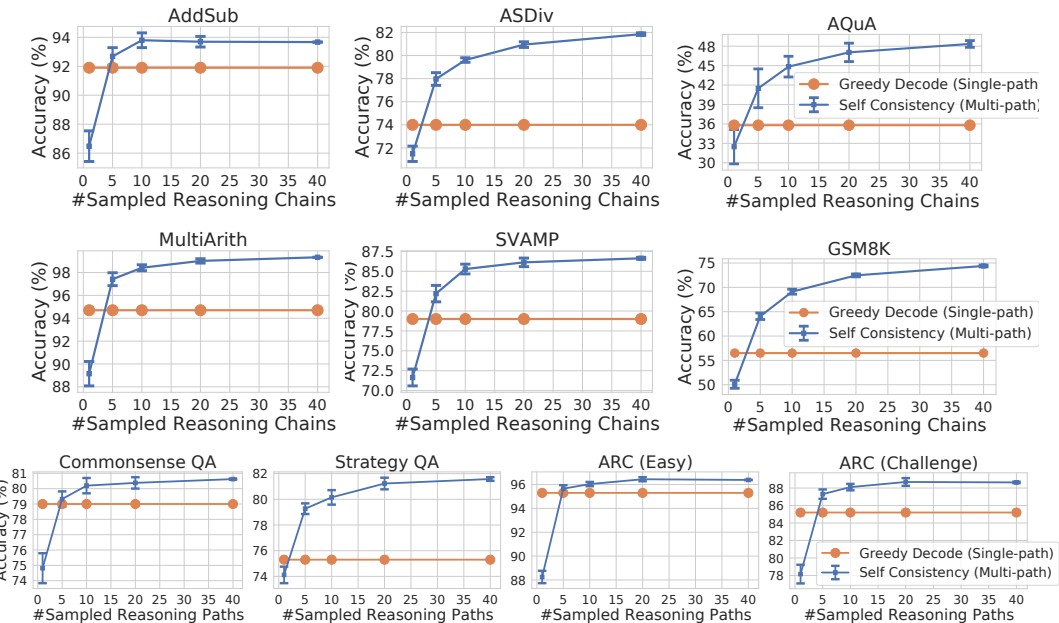

Figure 8: Self-consistency (blue) significantly improves accuracy across various arithmetic and commonsense reasoning tasks, over PaLM-540B. Sampling a higher number of diverse reasoning paths consistently helps reasoning accuracy.

| | Prompt set 1 (used in the main text) | Prompt set 2 | Prompt set 3 |
|---|---|---|---|
| CoT (Wei et al., 2022) | 56.5 | 54.6 | 54.0 |
| Self-consistency | 74.4 (+17.9) | 72.1 (+17.5) | 70.4 (+16.4) |

Table 9: GSM8K accuracy over PaLM-540B. The results show robustness of self-consistency with respect to different prompts in the input.

taking the majority vote (averaged over 10 runs). Note this is a typical ensemble approach (averaging over the predictions over multiple models) and it achieves a performance significantly worse than self-consistency (self-consistency over PaLM-540B gets an accuracy of 74.4%), as lower-capacity models drag down the performance of higher-capacity models. In addition, this approach is limited in two ways: 1) It requires multiple models for an ensemble which might not always be available, while self-consistency only requires one single model to "self-ensemble"; 2) If one of the models is much weaker, it can actually hurt the final performance.

| | Method | GSM8K accuracy |
|---|---|---|
| Single model | PaLM-540B, greedy / self-consistency | 56.5 / 74.4 |
| Ensemble of models | LaMDA-137B + PaLM-540B | $36.9 \pm 0.5$ |
| | PaLM-540B + GPT-3 (code-davinci-001, 175B) | $36.6 \pm 0.4$ |
| | LaMDA-137B + GPT-3 (code-davinci-001, 175B) | $16.0 \pm 0.8$ |
| | LaMDA-137B + PaLM-540B + GPT-3 (code-davinci-001, 175B) | $33.3 \pm 0.7$ |

Table 10: Comparison of GSM8K accuracy over multiple-model ensembles.

### A.1.4 COMBINING SELF-CONSISTENCY WITH OTHER ENSEMBLING STRATEGIES

Self-consistency is completely compatible with other ensemble strategies, although the gains achieved by self-consistency are significantly higher than other ensemble strategies (and can "override" the performance gains achieved by other ensemble strategies). We further performed experiments and include the results in Table 11 (for a fair comparison, we use 40 sets of prompts, or 40 prompt permutations to compare with self-consistency with 40 paths, all experiments are based on PaLM-540B).

|  | GSM8K accuracy |
|---|---|
| Different sets of prompts (x40) | 58.9 |
| Prompt-permutation (x40) | 59.6 |
| Self-consistency (x40) | 74.4 |
| Self-consistency + different sets of prompts (x40) | 75.4 |
| Self-consistency + prompt-permutation (x40) | 73.8 |

Table 11: Combining self-consistency with other ensembling strategies.

## A.2 DETAILS ON RESOURCES AND INFERENCE

For all four language models we perform prompting-based inference only. For UL2 we use TPU v3 (2x2 configuration, 4 chips, 8 cores). For GPT-3 models the experiments are done though the public API.[10] For LaMDA-137B we use TPU v3 (8x8 configuration, 64 chips, 128 cores). For PaLM-540B we use TPU v4 (4x4x12 configuration, 192 chips, 384 cores). Most inference jobs take 1 to 4 hours (over about 1,000 examples) for each task on UL2 and LaMDA-137B, and about 2 to 12 hours on PaLM-540B. Some tasks (e.g., commonsense reasoning) take longer but do not exceed 2 days for each task.

For GPT-3 models, we use 128 max tokens for all methods, without frequency penalty or presence penalty. For all models, we take the generated outputs until the start of the next "Q:" to parse the final answers, consistent with our prompting format.

| Question | Greedy Decode | Sampled Path 1 | Sampled Path 2 |
|---|---|---|---|
| **[MultiArith]** *Sam had 10 socks. If he threw away 3 old ones that didn't fit and bought 36 new ones, how many socks would he have?* | Sam had 10 socks. He threw away 3 old ones that didn't fit. He bought 36 new ones. That means he has 10 + 36 - 3 = 41 socks. The answer is 41. | Sam had 10 socks. He threw away 3 old ones. So now he has 7 left. Then he bought 36 new ones. So now he has 7 + 36 = 43 socks. The answer is 43. | He had 10 socks. He threw away 3 old ones that didn't fit. He bought 36 new ones. So in total he has 10 - 3 + 36 = 43 socks. The answer is 43. |
| **[SVAMP]** *In a school there are 697 girls and the rest are boys. If there are 228 more girls than boys. How many boys are there in that school?* | There are 697 girls and the rest are boys. That means there are 697 + 228 = 925 girls. The rest are boys. So the number of boys is 925 - 697 = 228. The answer is 228. | The number of boys is the difference between 697 and 228. This is 697 - 228 = 469. The answer is 469. | In the beginning there are 697 girls and the rest are boys. So, the number of boys must be 697 - 228 = 469 boys. The answer is 469. |
| **[AQuA]** *There are 10 oranges in a basket. Find the no. of ways in which 2 oranges are chosen from the basket?* (a) 45 (b) 90 (c) 120 (d) 150 (e) 180 | There are 10 oranges in a basket. So there are 10 ways in which 2 oranges are chosen from the basket. | There are 10 oranges in a basket. We need to choose 2 oranges from the basket. So the number of ways are 10C2. So the answer is (a). | There are 10 oranges in the basket. If you choose 2 oranges, you have 10C2 = 45 ways of choosing 2 oranges. So the answer is (a). |
| **[CommonsenseQA]** *The man laid on the soft moss and looked up at the trees, where was the man?* (a) niagra falls (b) forest (c) waterfall (d) ground (e) tree | The answer must be something that is soft and mossy. Of the above choices, only ground is soft and mossy. So the answer is (d). | The answer must be somewhere where moss is found. Of the above choices, only forests are known for having moss. So the answer is (b). | The answer must be a place where the man is laying down and looking up at trees. Of the above choices, only forest is a place where people lay down and look at trees. So the answer is (b). |
| **[ARC]** *Which of these will most likely harm a habitat?* (a) planting trees. (b) water pollution. (c) rainfall. (d) sunlight. | Planting trees will not harm a habitat. So the answer is (a). | The answer is (b), water pollution, because it will most likely harm the habitat. The other answers are not likely to harm the habitat. | The answer is (b), water pollution. Trees are good for a habitat, rainfall is necessary for a habitat, and sunlight is necessary for a habitat. So the answer is (b). |
| **[ARC-challenge]** *Which process in the carbon cycle takes the longest to complete?* (a) emission of waste. (b) respiration in animals. (c) photosynthesis in plants. (d) formation of fossil fuels. | The process of photosynthesis in plants takes the longest to complete. So the answer is (c). | The formation of fossil fuels takes the longest time to complete. So the answer is (d). | The process that takes the longest time to complete is the formation of fossil fuels, which happens over millions of years. So the answer is (d). |

Table 12: Additional examples where self-consistency helps repair the errors over greedy decode on LaMDA-137B. Two sampled reasoning paths that are consistent with the ground truth are shown.

---

[10] https://beta.openai.com/docs/api-reference/making-requests

| Question | Greedy Decode | Sampled Path 1 | Sampled Path 2 |
|---|---|---|---|
| **[GSM8K]** *Kyle bought last year's best-selling book for $19.50. This is with a 25% discount from the original price. What was the original price of the book?* | The book was sold for $19.50. This is 25% less than the original price. This means the original price was 100%. 25% of 100% is 25%. So the original price was 100% - 25% = 75%. The original price was 75% of $19.50. 75% of $19.50 is 75% * $19.50 = $14.625. The answer is $14.625. | The original price of the book is $19.50. Since it was discounted by 25%, the original price was 19.50 / 0.75 = 26. The answer is 26. | Kyle bought a book that is sold for $19.50. With a 25% discount, the original price must be higher. If we call the original price x, then the discounted price is 0.75x. Since this is $19.50, we can write 0.75x = 19.50. Solving this, we find that the original price is x = 19.50 / 0.75 = $26. The answer is $26. |
| **[AQuA]** *The capacity of a tank of dimensions (8 m x 6 m x 2.5 m) is* (a) 120 litres (b) 1200 litres (c) 12000 litres (d) 120000 litres (e) None of these | The volume of the tank is 8 m * 6 m * 2.5 m = 120 $m^3$. So the capacity of the tank is 120 $m^3$. So the answer is (c). | The volume of the tank is 8m * 6m* 2.5m = 120m$^3$. This is equal to 120000 litres. So the answer is (d) | The volume of the tank is 8 m * 6 m * 2.5 m = 120 m3. One cubic meter contains 1000 litres, so the tank contains 120 m3 * 1000 liters/m3 = 120,000 liters. So the answer is (d). |
| **[StrategyQA]** *Yes or no: Can the Great Depression be treated with Prozac?* | The Great Depression was an economic depression. Prozac is a drug used to treat depression. Thus, the Great Depression could be treated with Prozac. So the answer is yes. | Prozac is an anti-depressant medication. The Great Depression is not a disease. Thus, Prozac cannot treat the Great Depression. So the answer is no. | The Great Depression is a time period. Prozac is used to treat depression (the mental illness). Thus, the Great Depression can not be treated with Prozac. So the answer is no. |
| **[ARC-easy]** *Which source of energy is nonrenewable?* (a) coal. (b) sunlight. (c) water. (d) wood. | Coal, sunlight, and water are all renewable sources of energy. Wood is a nonrenewable source of energy. So the answer is (d). | A nonrenewable source of energy is one in which the source is not replenished in a short amount of time. Thus, the only option that is nonrenewable is coal. So the answer is (a). | Coal is a fossil fuel and is formed from dead organisms from the past. Coal cannot be produced in a short period of time, so it is considered a nonrenewable source of energy. So the answer is (a). |
| **[ARC-challenge]** *Students studied the revolution of Earth around the Sun. During which month will Florida have the most sunlight energy available?* (a) March. (b) June. (c) September. (d) December. | The Earth is closest to the Sun in December, hence Florida will have the most sunlight energy available in December. So the answer is (d). | Earth is closest to the Sun on June 21, the summer solstice. So the answer is (b). | Florida is closer to the sun during June, so it will have the most sunlight energy during that month. So the answer is (b). |

Table 13: Additional examples where self-consistency helps repair the errors over greedy decode on PaLM-540B. Two sampled reasoning paths that are consistent with the ground truth are shown.

## A.3 FULL SETS OF PROMPTS

We list the full details of the prompts used for two newly-introduced datasets, AQUA-RAT (Ling et al., 2017) and AI2 Reasoning Challenge (ARC) (Clark et al., 2018), where we manually composed the example chain-of-thought in this paper, in Table 14 and Table 15, respectively.

Table 14: Few-shot exemplars for AQUA-RAT.

**Q:** John found that the average of 15 numbers is 40. If 10 is added to each number then the mean of the numbers is? Answer Choices: (a) 50 (b) 45 (c) 65 (d) 78 (e) 64

**A:** If 10 is added to each number, then the mean of the numbers also increases by 10. So the new mean would be 50. The answer is (a).

**Q:** If a / b = 3/4 and 8a + 5b = 22,then find the value of a. Answer Choices: (a) 1/2 (b) 3/2 (c) 5/2 (d) 4/2 (e) 7/2

**A:** If a / b = 3/4, then b = 4a / 3. So 8a + 5(4a / 3) = 22. This simplifies to 8a + 20a / 3 = 22, which means 44a / 3 = 22. So a is equal to 3/2. The answer is (b).

**Q:** A person is traveling at 20 km/hr and reached his destiny in 2.5 hr then find the distance? Answer Choices: (a) 53 km (b) 55 km (c) 52 km (d) 60 km (e) 50 km

**A:** The distance that the person traveled would have been 20 km/hr * 2.5 hrs = 50 km. The answer is (e).

**Q:** How many keystrokes are needed to type the numbers from 1 to 500? Answer Choices: (a) 1156 (b) 1392 (c) 1480 (d) 1562 (e) 1788

**A:** There are 9 one-digit numbers from 1 to 9. There are 90 two-digit numbers from 10 to 99. There are 401 three-digit numbers from 100 to 500. 9 + 90(2) + 401(3) = 1392. The answer is (b).

Table 15: Few-shot exemplars for ARC easy/challenge.

**Q:** George wants to warm his hands quickly by rubbing them. Which skin surface will produce the most heat? (a) dry palms. (b) wet palms. (c) palms covered with oil. (d) palms covered with lotion.

**A:** Dry surfaces will more likely cause more friction via rubbing than other smoother surfaces, hence dry palms will produce the most heat. The answer is (a).

**Q:** Which factor will most likely cause a person to develop a fever? (a) a leg muscle relaxing after exercise. (b) a bacterial population in the bloodstream. (c) several viral particles on the skin. (d) carbohydrates being digested in the stomach.

**A:** Option (b), bacterial population is the most likely cause for a person developing fever. The answer is (b).

**Q:** Which change in the state of water particles causes the particles to become arranged in a fixed position? (a) boiling. (b) melting. (c) freezing. (d) evaporating.

**A:** When water is freezed, the particles are arranged in a fixed position; the particles are still moving for all other options. The answer is (c).

**Q:** When a switch is used in an electrical circuit, the switch can (a) cause the charge to build. (b) increase and decrease the voltage. (c) cause the current to change direction. (d) stop and start the flow of current.

**A:** The function of a switch is to start and stop the flow of a current. The answer is (d).

Table 16: Few-shot exemplars for HotpotQA (closed-book setting).

**Q:** Which magazine was started first Arthur's Magazine or First for Women?

**A:** Arthur's Magazine started in 1844. First for Women started in 1989. So Arthur's Magazine was started first. The answer is Arthur's Magazine.

**Q:** The Oberoi family is part of a hotel company that has a head office in what city?

**A:** The Oberoi family is part of the hotel company called The Oberoi Group. The Oberoi Group has its head office in Delhi. The answer is Delhi.

**Q:** What nationality was James Henry Miller's wife?

**A:** James Henry Miller's wife is June Miller. June Miller is an American. The answer is American.

**Q:** The Dutch-Belgian television series that "House of Anubis" was based on first aired in what year?

**A:** "House of Anubis" is based on the Dutch–Belgian television series Het Huis Anubis. Het Huis Anubis is first aired in September 2006. The answer is 2006.

As additional information, we also list the exact set of prompts used for all arithmetic reasoning tasks in Table 17, since there are multiple sets of prompts introduced in Wei et al. (2022). The prompts for CommonsenseQA and StrategyQA are the same as used in Wei et al. (2022).

We provide the exact prompts used for common NLP tasks in the following tables as well, including NLI (Table 18, Table 19, Table 20) and Closed-Book Question-Answering tasks (Table 16, Table 21).

Table 17: Few-shot exemplars for all arithmetic reasoning tasks, from Wei et al. (2022).

**Q:** There are 15 trees in the grove. Grove workers will plant trees in the grove today. After they are done, there will be 21 trees. How many trees did the grove workers plant today?

**A:** We start with 15 trees. Later we have 21 trees. The difference must be the number of trees they planted. So, they must have planted 21 - 15 = 6 trees. The answer is 6.

**Q:** If there are 3 cars in the parking lot and 2 more cars arrive, how many cars are in the parking lot?

**A:** There are 3 cars in the parking lot already. 2 more arrive. Now there are 3 + 2 = 5 cars. The answer is 5.

**Q:** Leah had 32 chocolates and her sister had 42. If they ate 35, how many pieces do they have left in total?

**A:** Leah had 32 chocolates and Leah's sister had 42. That means there were originally 32 + 42 = 74 chocolates. 35 have been eaten. So in total they still have 74 - 35 = 39 chocolates. The answer is 39.

**Q:** Jason had 20 lollipops. He gave Denny some lollipops. Now Jason has 12 lollipops. How many lollipops did Jason give to Denny?

**A:** Jason had 20 lollipops. Since he only has 12 now, he must have given the rest to Denny. The number of lollipops he has given to Denny must have been 20 - 12 = 8 lollipops. The answer is 8.

**Q:** Shawn has five toys. For Christmas, he got two toys each from his mom and dad. How many toys does he have now?

**A:** He has 5 toys. He got 2 from mom, so after that he has 5 + 2 = 7 toys. Then he got 2 more from dad, so in total he has 7 + 2 = 9 toys. The answer is 9.

**Q:** There were nine computers in the server room. Five more computers were installed each day, from monday to thursday. How many computers are now in the server room?

**A:** There are 4 days from monday to thursday. 5 computers were added each day. That means in total 4 * 5 = 20 computers were added. There were 9 computers in the beginning, so now there are 9 + 20 = 29 computers. The answer is 29.

**Q:** Michael had 58 golf balls. On tuesday, he lost 23 golf balls. On wednesday, he lost 2 more. How many golf balls did he have at the end of wednesday?

**A:** Michael initially had 58 balls. He lost 23 on Tuesday, so after that he has 58 - 23 = 35 balls. On Wednesday he lost 2 more so now he has 35 - 2 = 33 balls. The answer is 33.

**Q:** Olivia has $23. She bought five bagels for $3 each. How much money does she have left?

**A:** She bought 5 bagels for $3 each. This means she spent 5 * $3 = $15 on the bagels. She had $23 in beginning, so now she has $23 - $15 = $8. The answer is 8.

Table 18:  Few-shot exemplars for ANLI.

Premise:
"Conceptually cream skimming has two basic dimensions - product and geography."
Based on this premise, can we conclude the hypothesis "Product and geography are what make cream skimming work." is true?
OPTIONS:
- yes
- no
- it is not possible to tell

**A:** Based on "cream skimming has two basic dimensions" we can't infer that these two dimensions are what make cream skimming work. The answer is it is not possible to tell.

Premise:
"One of our member will carry out your instructions minutely."
Based on this premise, can we conclude the hypothesis "A member of my team will execute your orders with immense precision." is true?
OPTIONS:
- yes
- no
- it is not possible to tell

**A:** "one of" means the same as "a member of", "carry out" means the same as "execute", and "minutely" means the same as "immense precision". The answer is yes.

Premise:
"Fun for adults and children."
Based on this premise, can we conclude the hypothesis "Fun for only children." is true?
OPTIONS:
- yes
- no
- it is not possible to tell

**A:** "adults and children" contradicts "only children". The answer is no.

Premise:
"He turned and smiled at Vrenna."
Based on this premise, can we conclude the hypothesis "He smiled at Vrenna who was walking slowly behind him with her mother." is true?
OPTIONS:
- yes
- no
- it is not possible to tell

**A:** the premise does not say anything about "Vrenna was walking". The answer is it is not possible to tell.

Premise:
"well you see that on television also"
Based on this premise, can we conclude the hypothesis "You can see that on television, as well." is true?
OPTIONS:
- yes
- no
- it is not possible to tell

**A:** "also" and "as well" mean the same thing. The answer is yes.

Premise:
"Vrenna and I both fought him and he nearly took us."
Based on this premise, can we conclude the hypothesis "Neither Vrenna nor myself have ever fought him." is true?
OPTIONS:
- yes
- no
- it is not possible to tell

**A:** "Vrenna and I both" contradicts "neither Vrenna nor myself". The answer is no.

Table 19: Few-shot exemplars for e-SNLI (the rationales are crowd-sourced annotations from the original dataset).

Premise:
"A person on a horse jumps over a broken down airplane."
Based on this premise, can we conclude the hypothesis "A person is training his horse for a competition." is true?
OPTIONS:
- yes
- no
- it is not possible to tell
**A:** The person is not necessarily training his horse. The answer is it is not possible to tell.

Premise:
"A person on a horse jumps over a broken down airplane."
Based on this premise, can we conclude the hypothesis "A person is at a diner, ordering an omelette." is true?
OPTIONS:
- yes
- no
- it is not possible to tell
**A:** One jumping horse cannot be in a diner ordering food. The answer is no.

Premise:
"A person on a horse jumps over a broken down airplane."
Based on this premise, can we conclude the hypothesis "A person is outdoors, on a horse." is true?
OPTIONS:
- yes
- no
- it is not possible to tell
**A:** A broken down airplane is outdoors. The answer is yes.

Premise:
"Children smiling and waving at camera."
Based on this premise, can we conclude the hypothesis "They are smiling at their parents." is true?
OPTIONS:
- yes
- no
- it is not possible to tell
**A:** Just because they are smiling and waving at a camera does not imply their parents or anyone is anyone behind it. The answer is it is not possible to tell.

Premise:
"Children smiling and waving at camera."
Based on this premise, can we conclude the hypothesis "The kids are frowning." is true? OPTIONS:
- yes
- no
- it is not possible to tell
**A:** One cannot be smiling and frowning at the same time. The answer is no.

Premise:
"Children smiling and waving at camera."
Based on this premise, can we conclude the hypothesis "There are children present." is true?
OPTIONS:
- yes
- no
- it is not possible to tell
**A:**The children must be present to see them smiling and waving. The answer is yes.

Table 20: Few-shot exemplars for RTE.

Premise:
"No Weapons of Mass Destruction Found in Iraq Yet."
Based on this premise, can we conclude the hypothesis "Weapons of Mass Destruction Found in Iraq." is true?

**A:** "No Weapons of Mass Destruction Found" contradicts "Weapons of Mass Destruction Found". The answer is no.

Premise:
"A place of sorrow, after Pope John Paul II died, became a place of celebration, as Roman Catholic faithful gathered in downtown Chicago to mark the installation of new Pope Benedict XVI."
Based on this premise, can we conclude the hypothesis "Pope Benedict XVI is the new leader of the Roman Catholic Church." is true?'

**A:** "installation of new Pope Benedict XVI." means "Pope Benedict XVI is the new leader". The answer is yes.

Premise:
"A man is due in court later charged with the murder 26 years ago of a teenager whose case was the first to be featured on BBC One's Crimewatch. Colette Aram, 16, was walking to her boyfriend's house in Keyworth, Nottinghamshire, on 30 October 1983 when she disappeared. Her body was later found in a field close to her home. Paul Stewart Hutchinson, 50, has been charged with murder and is due before Nottingham magistrates later."
Based on this premise, can we conclude the hypothesis "Paul Stewart Hutchinson is accused of having stabbed a girl." is true?

**A:** The premise does not say Paul Stewart Hutchinson "stabbed" this girl. The answer is no.

Premise:
"Herceptin was already approved to treat the sickest breast cancer patients, and the company said, Monday, it will discuss with federal regulators the possibility of prescribing the drug for more breast cancer patients."
Based on this premise, can we conclude the hypothesis "Herceptin can be used to treat breast cancer." is true?

**A:** "Herceptin was approved to treat breast cancer" implies that "Herceptin can be used to treat breast cancer". The answer is yes.

Table 21: Few-shot exemplars for BoolQ (closed-book setting).

**Q:** does system of a down have 2 singers?

**A:** System of a Down currently consists of Serj Tankian, Daron Malakian, Shavo Odadjian and John Dolmayan. Serj and Daron do vocals, so the band does have two singers. The answer is yes.

**Q:** do iran and afghanistan speak the same language?

**A:** Iran and Afghanistan both speak the Indo-European language Persian. The answer is yes.

**Q:** is a cello and a bass the same thing?

**A:** The cello is played sitting down with the instrument between the knees, whereas the double bass is played standing or sitting on a stool. The answer is no.

**Q:** can you use oyster card at epsom station?

**A:** Epsom railway station serves the town of Epsom in Surrey and is not in the London Oyster card zone. The answer is no.

