# OpenReview forum: "Self-Consistency Improves Chain of Thought Reasoning in Language Models"
_ICLR.cc/2023/Conference — ICLR 2023 poster_

### Official Review · Reviewer_Tj1M · 2022-10-23

**Confidence:** 4
**Clarity, Quality, Novelty And Reproducibility:** Please refer to Strength And Weaknesses.
**Correctness:** 3
**Technical Novelty And Significance:** 2
**Empirical Novelty And Significance:** 3
**Recommendation:** 5

**Strength And Weaknesses:**

Strength:
1 Motivation is straightforward.
2 The method is simple and effective.
3 The author conducted a variety of experiments to assess the method’s performance.

Weak:
Self-consistency is an interesting point. However, my main consideration is that this paper does not go deep enough into the study of self-consistency. This lack of depth is mainly reflected in two perspectives (1) only the majority vote of multiple randomly generated answers is considered. There is a lack of in-depth observation on how to combine different answers; (2) different answers are obtained by controlling the temperature and k. For text generation, the effect of these two hyperparameters is unclear. In particular, how these two hyperparameters affect answer consistency is unclear.

Other weak points:
1 Experiment results suggest that LM cannot distinguish well between correct and incorrect answers (at least from the aspect of token probability), but why taking majority vote over answers generated from different reasoning paths works? More in-depth studies should be done.
2 Since “diversity of reasoning paths is the key to better performance”, why other ensemble-based approaches (e.g. the multiple prompts setting) didn’t get larger gain for their diversity?
3 Figure 2 is confusing: why greedy decode has different number of sampled paths?
4 It would be better to show full sampled paths in table 4 and other examples if the purpose is to claim self-consistency can repair errored reasoning process.



**Summary Of The Paper:**

This paper proposed a new decoding and answer generation method for LMs to achieve multi-step math problems in a “step-by-step” way. Different from the standard “chain-of-thought” prompting which uses greedy decoding, this method samples several reasoning paths generated by LM, and then generates the final answer by finding the most frequent answer. Author conducted a series of experiments varying in LM type and LM size to assess the self-consistency method’s performance.

**Summary Of The Review:**

Overall, I suggest that the authors do a deeper study of the problem, e.g., for LMs, how does the problem of consistency arise and what is its fundamental nature? What exactly does improving self-consistency mean? Why does majority vote improve the model effect?

---

> ### Author Response · Authors · 2022-11-16
> **Response to reviewer Tj1M**
>
> Thank you for the detailed review!
>
> > (1) only the majority vote of multiple randomly generated answers is considered. There is a lack of in-depth observation on how to combine different answers
>
> Please note that we have experimented with **5** answer aggregation strategies (see Table 1 in Section 2 for an in-depth study of different strategies), including weighted average and weighted sum, each with both normalized and unnormalized log probabilities, as well as unweighted sum (equivalent to taking a majority vote). As discussed in Section 2, taking the weighted sum is the exact way to compute the probability of answer $a$ given by the sampling and marginalization process (i.e., Equation 1), and it is consistent with our experiment results (Table 1) which shows among the 5 methods, the weighted sum (with normalized log probability) achieves the best performance on various reasoning tasks. Further, we also explained that taking an unweighted sum (i.e., majority vote) is an approximation of computing the exact probability in Equation 1, and this is due to the language models are not well-calibrated and thus the normalized log probabilities of each generation are quite close to each other.
>
>
> > (2) different answers are obtained by controlling the temperature and k. For text generation, the effect of these two hyperparameters is unclear. In particular, how these two hyperparameters affect answer consistency is unclear.
>
> We actually have the ablation experiments in Figure 4, showing the effect of the temperature $T$ and $k$, as well as $p$ in nucleus sampling. From the results we can see that self-consistency is in general quite robust to different choices of sampling strategies and parameters, as long as there is enough diversity introduced in the sampling process (for example, if you use a much lower temperature $T=0.3$, which is closer to greedy decoding with a lower diversity, then the performance degrades a little bit compared to higher values of $T$).
>
>
>
> > 1 Experiment results suggest that LM cannot distinguish well between correct and incorrect answers (at least from the aspect of token probability), but why taking majority vote over answers generated from different reasoning paths works? More in-depth studies should be done.
>
> As we already discussed in the introduction section, “Self-consistency leverages the intuition that complex reasoning tasks typically admit multiple reasoning paths that reach a correct answer. The more that deliberate thinking and analysis is required for a problem, the greater the diversity of reasoning paths that can recover the answer. ”
>
> As an analogy, this process is similar to how a human might solve a difficult reasoning task. For example, assume a human is solving a math task, they can come up with 2 different ways to solve it, where each can lead to a different final answer and they might have great confidence in both solutions (similar to how a language model assigns high log probability to each generation). But since the final answers are different, one of them must be wrong. Hence one possible way is to come up with a 3rd way of solving this problem, and check whether its answer is the same as the 1st or the 2nd solution. This explains why both correct and incorrect solutions might have high probability, but taking a majority vote (given different reasoning paths) could potentially give you a better final answer.
>
>
> > 2 Since “diversity of reasoning paths is the key to better performance”, why other ensemble-based approaches (e.g. the multiple prompts setting) didn’t get larger gain for their diversity?
>
> We found that multiple prompts setting (varying the input space) yields a much lower diversity in the reasoning paths compared to sampling in the decoder (varying in the output space). This is potentially due to the nature of how language models work, as language models could have a strong prior that small perturbations in the input space do not cause a big difference in the outputs, while direct sampling in the output space is a much more effective way to explore various knowledge language models embed in their parameter space.
>
>
> > 3 Figure 2 is confusing: why greedy decode has different number of sampled paths?
>
> Sorry for the confusion, greedy decode doesn’t have a different number of sampled paths (that’s why it’s a flat line). It’s only shown as a baseline to compare with self-consistency with different number of paths.
>
>
> > 4 It would be better to show full sampled paths in table 4 and other examples if the purpose is to claim self-consistency can repair errored reasoning process.
>
> Thanks for the suggestion! We will add more sampled paths in the main text. We currently have 2 sampled paths in Table 4 due to space constraints and we have shown additional examples and sampled paths in Appendix (Table 12 and 13). We have also attached the full sampled paths (all 40 of them) in supplementary materials so feel free to check the full results in the zip file.

---

> > ### Comment · Reviewer_Tj1M · 2022-12-01
> > **response**
> >
> > Thanks for your responses. However, I am not convinced by them. The self-consistency provided by adjusting the temperature and k may be unlikely to advance our understanding of self-consistency and chains-of-thought. Therefore, I would like to keep the rating.

---

### Official Review · Reviewer_Z6Ub · 2022-10-24

**Confidence:** 4
**Correctness:** 3
**Technical Novelty And Significance:** 3
**Empirical Novelty And Significance:** 4
**Recommendation:** 6

**Clarity, Quality, Novelty And Reproducibility:**

The paper is well-written overall and its motivation is quite clear i.e. improving recently proposed CoT. The claims are supported by extensive experiments with a variety of model configurations. I also do not have any major concerns about reproducibility since the authors provide sufficient descriptions and artifacts (code, prompts, model outputs, etc).

In terms of originality, sampling different model outputs and ranking them according to a metric has been largely explored in the past (e.g. MT, text generation). In the context of pre-trained LMs, the GPT3 paper used beam search results for free-form completion, Adiwardana et al., (2020) used a very similar sample-and-rank method based on regular sampling, and more recently Jung et al. (2022) used nucleus sampling to sample diverse generated outputs to improve reasoning [1]. The last one is arguably parallel work but I am surprised it was not cited.

Having said that, the approach itself is not that novel but it is a straightforward direction to explore for CoT LMs and the community can benefit from such a simple method and large-scale empirical findings.

Questions:
- It looks like that self-consistency is an instance of the sample-and-rank method that uses nucleus sampling instead of temperature sampling and majority vote (simple sum) instead of log probability. Do the authors share this view? I'd suggest making the connection clear to avoid misinterpretations regarding novelty.
- In table 1, could the authors specify on which set the accuracy scores are computed? It's likely not very crucial since the majority vote was eventually selected, but it would be useful to know.

[1] https://arxiv.org/pdf/2205.11822.pdf

**Strength And Weaknesses:**

**Strengths**

Proposes a simple and effective decoding strategy to improve LMs with CoT prompting.

Contains extensive experiments with different LM types and sizes that demonstrate large gains compared to vanilla CoT without requiring additional training.

Self-consistency strategy surfaces diverse outputs that correspond to rationales in CoT-prompted LMs and improves robustness to imperfect inputs. Also, it can provide uncertainty estimates for calibration.

**Weaknesses**
The decoding strategy itself has limited novelty since sampling strategies (including nucleus sampling) and ranking according to model scores have been previously used for pre-trained LMs albeit not with CoT prompting.

Even though it generally leads to improvements, the proposed decoding strategy creates a significant overhead proportional to the number of outputs sampled e.g 5x latency for 5 samples assuming that samples have equal lengths.

The marginalization process of self-consistency assumes that answers are coming from a fixed answer set, which decreases the scope of applicability.

**Summary Of The Paper:**

This paper proposes a decoding strategy to improve chain-of-thought (CoT) prompting in pre-trained language models. The proposed strategy uses nucleus sampling to sample diverse outputs as opposed to a single one in CoT (based on greedy decoding) and selects the answer that is most consistent through marginalization. Evaluation performed on arithmetic and commonsense reasoning shows large improvements compared to CoT. Results are consistent across different language models and sizes.

**Summary Of The Review:**

Overall, this paper conducts a large empirical study of a simple decoding strategy for pre-trained LMs that use CoT prompting. The decoding strategy itself has limited novelty but the empirical findings are quite interesting/useful. The experiments show strong improvements over vanilla CoT prompting, which should be of interest to the community.

---

> ### Author Response · Authors · 2022-11-16
> **Response to reviewer Z6Ub**
>
> Thanks for your feedback! We will add citations to Jung et al. (2022) and add more discussions on the novelty of our method.
>
> > It looks like that self-consistency is an instance of the sample-and-rank method that uses nucleus sampling instead of temperature sampling and majority vote (simple sum) instead of log probability.
>
> Sorry for the confusion, self-consistency is actually different from the sample-and-rank method, please see a comparison between our method and the sample-and-rank method (Adiwardana et al., (2020)) in Figure 3 of Section 3.4, where we show that self-consistency can significantly outperform sample-and-rank. We also show that self-consistency outperforms beam search in Table 6 of Section 3.4.
>
> - For sampling, self-consistency is not specific to nucleus sampling, any standard sampling techniques can be used, including temperature sampling, top-k sampling, and nucleus sampling (please see the 2nd paragraph of Section 2). We also show ablation experiments by using different sampling strategies and found that self-consistency is quite robust to different sampling methods and parameters, please see Figure 4 (left) for a comparison on different sampling methods.
>
> - To explain the main differences in more detail, sample-and-rank assumes that a language model can rank different sampled generations quite well (based on its own log likelihood) and thus taking the top-ranked sequence would yield a better result (similarly for beam search). On the contrary, self-consistency does not assume that models can rank their generations well, in fact we found that on most reasoning tasks, the language models are not well-calibrated and thus their log likelihood can not distinguish between good and bad generations well (see footnote 2). Hence instead, we use CoT prompting to sample multiple reasoning paths (so the main point of sampling is to introduce a diverse set of reasoning paths) and taking a vote on the final answers (via marginalizing over the sampled reasoning paths). The intuition being a complex reasoning problem typically admits multiple different ways of thinking leading to its unique correct answer. For example, think about how a human might solve a complex math problem: the human could come up with 3 different ways of solving it, and they would be more confident on the final answer if it is the majority of the 3 answers, while sample-and-rank asks the human to rank the 3 ways by their confidence of each generation (and a human could be equally confident since they come up with the 3 generations independently). This sampling and marginalizing procedure is significantly different from the existing sample-and-rank procedure, and is not explored in existing literature. In addition, we also show a much larger improvement by self-consistency compared to existing sample-and-rank in Figure 3, showing the superiority of self-consistency over sample-and-rank with CoT-path-sampling and aggregation (instead of answer-sampling and ranking).
>
>
> > In table 1, could the authors specify on which set the accuracy scores are computed? It's likely not very crucial since the majority vote was eventually selected, but it would be useful to know.
>
> Sorry for the confusion, the experiments are done on the **test set** of all tasks (same as all experiments in the experiment section). As detailed in footnote 3, by default we use the test split of all datasets if the test split is available for evaluation. For CommonsenseQA we use the dev split since the test split is not available. We will add this detail in the final version.

---

### Official Review · Reviewer_s5hD · 2022-10-24

**Confidence:** 3
**Correctness:** 4
**Technical Novelty And Significance:** 2
**Empirical Novelty And Significance:** 4
**Recommendation:** 6

**Clarity, Quality, Novelty And Reproducibility:**

For clarity and quality see the previous section.
In terms of novelty, I think though authors self consistency method is indeed novel when applied on chain of thought prompting, the idea itself is somewhat an obvious next step and something that should work well. So even though the empirical results are extensive and solid, they are not as surprising. So I think the novelty of this paper is only marginal. However the empirical results are nevertheless significant.


**Strength And Weaknesses:**

Strength:
1.	The self-consistency method is very well motivated, and authors did a great job presenting this motivation. The method is very principled.
2.	The empirical evidence of the superiority of the method is extensive and solid.

Weakness: authors’ notation of r_i is a bit unclear to me, my understanding is that they are just the generated text before the final answer (what authors call reasoning path). It is not clear to me what is the difference between the generated text before the final answer and what authors call the reasoning paths. Did authors do something specific to exclude some potentially irrelevant text that were generated but did not contribute to reasoning? If not, perhaps calling them reasoning paths is a bit confusing? It seems r_i and a_i always appear together, which gives me the impression that the only distinction is that a_i is the final answer and r_i is everything before that.

Another confusion is the presentation of the weighted sum, is it correct to understand the probability weight is just the logsumexp of the logprobs of the generation? That makes sense to be, equations 1 is a bit misleading to me because it seems to place an emphasis on r_i as being the reasoning path when it’s just the probability of a specific generation. I think it is okay to mention the generation is supposed to be reasoning paths, but it might be more clear to just note them as generated text in those formal expressions.


**Summary Of The Paper:**


This paper investigates an improved version of Chain-of-thought prompting by leveraging self consistency and shows empirical evidence that their self consistency prompting indeed improve reasoning tasks performance compared with the original chain-of-thoguht prompting.


**Summary Of The Review:**

I’d recommend marginal acceptance, as I think the self consistency method is very principled and well motivated and empirical evidence is extensive and solid. However the notation and the presentation could be improved.

---

> ### Author Response · Authors · 2022-11-16
> **Response to reviewer s5hD**
>
> Thank you for your detailed review and acknowledging our method as “very principled and well motivated”. We will improve the notations and presentations of our method.
>
> > authors’ notation of r_i is a bit unclear to me, my understanding is that they are just the generated text before the final answer (what authors call reasoning path).
>
> Sorry for the confusion, we will make the notations more clear in the final version. Your understanding is correct that $a_i$ is the final answer and $r_i$ is everything before that. By using chain-of-thought prompting, where the few-shot exemplars follow the exact format of:
>
> Q: [question 1]
>
> A: [reasoning path 1]. The answer is [final answer 1].
>
> Q: [question 2]
>
> A: [reasoning path 2]. The answer is [final answer 2].
>
> …
>
> Q: [new question]
>
> A:
>
> Then the model follows this format to generate:
>
> **[$r_i$]. The answer is [$a_i$].**
>
> Here it is usually the case that $r_i$ is exactly the reasoning path and there’s rarely any irrelevant text generated in $r_i$ that does not contribute to the reasoning (based on our analysis on 100 examples manually). This is why we call $r_i$ the “reasoning path” but we will make the notations more clear.
>
> As examples, please see the exact model outputs in Table 4 (as well as more examples in Table 12 and 13 for two different models), where the models outputs always follow the format of "*[$r_i$]. The answer is [$a_i$].*" where $r_i$ is exactly the reasoning path.
>
> > Another confusion is the presentation of the weighted sum, is it correct to understand the probability weight is just the logsumexp of the logprobs of the generation? I think it is okay to mention the generation is supposed to be reasoning paths, but it might be more clear to just note them as generated text in those formal expressions.
>
> Thanks for the suggestion, we will make the notations more clear in those formal expressions. Yes, $r_i$ is indeed the generated text that is supposed to be the reasoning path. We put an emphasis on $r_i$ because we want to highlight that the marginalization is applied over $r_i$’s, i.e., after aggregation, the probability of each answer $a$ is the logsumexp of the logprobs of the generation of $r_i, a_i$, conditioned on the final answer $a_i$ being equal to $a$, regardless of the generated $r_i$’s before $a_i$ (as $r_i$’s could be very different, but they can all lead to the same $a_i$). We will make it clear that $r_i, a_i$ are both generated texts in those formal expressions.

---

### Official Review · Reviewer_avbc · 2022-10-25

**Confidence:** 5
**Clarity, Quality, Novelty And Reproducibility:** Refer to main review.
**Correctness:** 4
**Technical Novelty And Significance:** 3
**Empirical Novelty And Significance:** 4
**Recommendation:** 10

**Strength And Weaknesses:**

- **Clarity:** The paper is very clearly written and is easy to understand. The takeaways from each section are clear and flow from one to the other.
- The tasks, baseline sampling techniques, models and various tradeoffs while using different hyperparameters have been explored well! They give a clear picture of what is useful in a practical setting.
- **********************************Reproducibility:********************************** The method is easy to implement, but not all the models are publicly available or are accessible only through a paid API.
- The method itself is quite general and can be used across several task types and models.
- Further Discussion: CoT/ Scratchpads are useful in understanding the output of a model; how would the authors suggest choosing a reasoning path to provide an explanation for the answer given by the model?
- ******************************Novelty:****************************** The paper is novel in its approach.
- ******************Quality:****************** The paper, analysis and writing are of a very high quality

**Summary Of The Paper:**

The paper proposes a new way to improve the reasoning power of language models by decoding several sequences of CoT/Scratchpad reasoning and then using a voting-based scheme to choose an answer. At the cost of having to sample multiple times from the model, the authors show a consistent improvement across tasks and models. Through thorough analysis, the authors show the utility of their method, and its advantages over other sampling techniques and highlight the best practices that might be useful in practical applications.

**Summary Of The Review:**

The paper is extremely strong in almost exhaustively testing different settings and tasks. It has very limited weaknesses and will probably have a large impact on various practical applications.

---

> ### Author Response · Authors · 2022-11-16
> **Response to reviewer avbc**
>
> Thank you for the positive review! We are very encouraged that you found our method to be novel and general. We will add more discussions on how to choose a reasoning path: in practice if users just look for a single explanation for each question, then a random selection of the reasoning paths which has a consistent answer with the majority vote can be made; if a user is interested in more-thorough explanations and the possible different ways models can reach a certain decision, e.g., for learning purposes in math reasoning tasks, or for exploring what knowledge is stored in LLM and how it can be composed in commonsenseQA tasks, then we can show a diverse set of reasoning paths to provide a more comprehensive picture (e.g., via clustering all the reasoning paths with a consistent answer as the majority vote).

---

### Decision · Program_Chairs · 2023-01-20

**Decision:**

Accept: poster

**Justification For Why Not Higher Score:**

See meta-review

**Justification For Why Not Lower Score:**

If the outcome of a rejection is just that the paper is re-submitted, and four more reviewers + 1 AC weigh in (all of whom sit somewhere else on the "novelty vs. results" importance spectrum), I'm not sure it's productive to reject this paper.

**Metareview: Summary, Strengths And Weaknesses:**

The authors introduce an expansion of the chain of thought prompting framework. In this change, they generate multiple reasoning chains for each input example (using sampling methods) and explore different methods for aggregating the confidence in these reasoning chains to make a final prediction. In the end, using a majority vote among the predicted final answer options of the reasoning chains seems to work best.

Strengths:

The authors all highlight the rigour of the evaluations

Weaknesses:

Most reviewers complained about the novelty. While I typically avoid critiquing novelty, this case is challenging. The paper tests a rigorous set of hypotheses and achieves a strong empirical improvement with the final approach, but the methodological question seems to simplify down to greedy decoding vs. sample + aggregate. The novelty of sample + aggregate is specific to CoT because there happens to be a correct / incorrect answer at the end of the reasoning chain. In a typical text generation paper, though, (and CoT is a form of text generation), this could potentially be explored as a hyperparameter in the initial task paper. As a result, the contribution seems a bit incremental. That being said, "conceptual novelty vs. impact of results" is a question of opinion, and the empirical improvements are indeed substantial, so I'll vote Accept.

**Note From Pc:**

if the above contains the word "oral" or "spotlight" please see: "oral" presentation means -> notable-top-5% and "spotlight" means -> notable-top-25%. As stated in our emails, we are disassociating presentation type from AC recommendations

**Summary Of Ac-Reviewer Meeting:**

N/A